# In Vitro Susceptibility of Aztreonam-Vaborbactam, Aztreonam-Relebactam and Aztreonam-Avibactam Associations against Metallo-β-Lactamase-Producing Gram-Negative Bacteria

**DOI:** 10.3390/antibiotics12101493

**Published:** 2023-09-29

**Authors:** Cécile Emeraud, Sandrine Bernabeu, Laurent Dortet

**Affiliations:** 1Department of Bacteriology-Hygiene, Bicêtre Hospital, Assistance Publique—Hôpitaux de Paris, 94270 Le Kremlin-Bicêtre, France; cecile.emeraud@aphp.fr (C.E.); sandrine.bernabeu@aphp.fr (S.B.); 2INSERM UMR 1184, RESIST Unit, Faculty of Medicine, Paris-Saclay University, 94270 Le Kremlin-Bicêtre, France

**Keywords:** metallo-β-lactamase, aztreonam, β-lactamase inhibitors

## Abstract

Background: Despite the availability of new options (ceftazidime-avibactam, imipenem-relebactam, meropenem-vaborbactam and cefiderocol), it is still very difficult to treat infections caused by metallo-β-lactamase (MBLs)-producers resistant to aztreonam. The in vitro efficacy of aztreonam in association with avibactam, vaborbactam or relebactam was evaluated on a collection of MBL-producing Enterobacterales, MBL-producing *P. aeruginosa* and highly drug-resistant *S. maltophilia*. Methods: A total of fifty-two non-duplicate MBL-producing Enterobacterales, five MBL-producing *P. aeruginosa* and five multidrug-resistant *S. maltophila* isolates were used in this study. The minimum inhibitory concentrations (MICs) of aztreonam, meropenem-vaborbactam and imipenem-relebactam were determined by Etest^®^ (bioMérieux, La Balme-les-Grottes) according to EUCAST recommendations. For aztreonam-avibactam, aztreonam-vaborbactam and aztreonam-relebactam associations, the MICs were determined using Etest^®^ on Mueller-Hinton (MH) agar supplemented with 8 mg/L of avibactam, 8 mg/L of vaborbactam and 4 mg/L of relebactam. The MICs were interpreted according to EUCAST guidelines. Results: The susceptibility rates of aztreonam-avibactam, aztreonam-vaborbactam and aztreonam-relebactam with a standard exposure of aztreonam (1g × 3, IV) were 84.6% (44/52), 55.8% and 34.6% for Enterobacterales and 0% for all combinations for *P. aeruginosa* and *S. maltophila*. The susceptibility rates of aztreonam-avibactam, aztreonam-vaborbactam and aztreonam-relebactam with a high exposure of aztreonam (2g × 4, IV) were 92.3%, 78.9% and 57.7% for Enterobacterales, 75%, 60% and 60% for *P. aeruginosa* and 100%, 100% and 40% for *S. maltophila*. Conclusions: As previously demonstrated for an aztreonam/ceftazidime-avibactam combination, aztreonam plus imipenem-relebactam and aztreonam plus meropenem-vaborbactam might be useful options, but with potentially lower efficiency, to treat infections caused by aztreonam-non-susceptible MBL-producing Gram-negative strains.

## 1. Introduction

The extensive dissemination of carbapenemase-producing Gram-negative bacteria poses a global threat to public health. It is crucial to implement new therapeutic strategies to treat infections caused by highly resistant pathogens. Carbapenemase belong to three of four classes of the Ambler classification: class A carbapenemases (mostly KPC types) [1], class B carbapenemases or metallo-β-lactamases (MBLs) (mostly NDM, VIM, or IMP types) [2], and class D carbapenemases (mostly OXA-48-like types in Enterobacterales) [3]. Recently, new therapeutic alternatives have been marked for the treatment of carbapenemase producers. These new drugs include ceftazidime-avibactam, meropenem-vaborbactam, imipenem-relebactam and cefiderocol. Despite the availability of these new options, it is still very difficult to treat infections caused by MBL producers resistant to aztreonam. Indeed, these isolates are resistant to ceftazidime-avibactam, meropenem-vaborbactam and imipenem-relebactam through the MBL-hydrolyzing activity, which is not inhibited by the added β-lactamase inhibitor. Regarding cefiderocol, it might be active on MBLs, but NDM producers have cefiderocol’s MIC_50_ close to the clinical breakpoint of 2 mg/L, resulting in only 70% susceptibility [4]. Since current MBLs, such as VIM, NDM or IMP, are not able to hydrolyze aztreonam, resistance to this antimicrobial in MBL producers is due to the co-production of an ESBL or an AmpC. Accordingly, it has been demonstrated that the concomitant use of aztreonam plus ceftazidime-avibactam or amoxicillin-clavulanate or ceftolozane-tazobactam are accurate options to treat infections caused by MBL + ESBL or MBL + AmpC producers, with aztreonam plus ceftazidime-avibactam being the most powerful strategy [5]. Unfortunately, is has been recently shown that aztreonam-avibactam resistance emerged in NDM-producing *E. coli* through the association of modified penicillin-binding protein 3 (PBP3) and the production of CMY-42 cephalosporinase [6]. Thus, it is crucial to consider other therapeutic alternatives. In this study, we evaluated the susceptibility of aztreonam-resistant MBL-producing isolates to aztreonam in association with two other marketed β-lactamase inhibitors: vaborbactam and relebactam. Vaborbactam, a cyclic boronic β-lactamase inhibitor and relebactam, a diazabicyclooctane inhibitor, are very potent against KPC enzymes, but are also active on diverse class A (including ESBL) and class C β-lactamases (AmpC) [7]. As previously demonstrated for aztreonam-avibactam [5], aztreonam-vaborbactam and aztreonam-relebactam seemed to be effective on NDM-producing *K. pneumoniae* [8] and few multidrug-resistant *S. maltophila* [9]. Here, we assessed the susceptibility of aztreonam in combination with vaborbactam or relebactam or avibactam on a collection of MBL producers including Enterobacterales (including species other than *K. pneumoniae*), *P. aeruginosa* and *S. maltophilia*.

## 2. Results

A total of fifty-two non-duplicate MBL-producing Enterobacterales, five MBL-producing *P. aeruginosa* and five multidrug-resistant *S. maltophila* isolates were used in this study. These isolates have already been tested for aztreonam-avibactam in a previous study [5]. All strains were resistant to aztreonam due to the co-production of an ESBL and/or a cephalosporinase. The MBL-producing Enterobacterales included 30 NDM-producers with 11 isolates co-producing one OXA-48-like carbapenemase, fourteen VIM-producers, six IMP-producers, one GIM-1-producer and one TMB-1-producer. The MBL-producing *P. aeruginosa* were one VIM-2-, three IMP-1- and one IMP-2-producers. *S. maltophila* isolates were resistant to all β-lactams including ticarcillin-clavulanate, and to all other antimicrobials including fluoroquinolones and trimethoprim-sulfamethoxazole [5].

None of the strains included in the study were sensitive to aztreonam at standard exposure (the MICs range from 2 to >256 mg/L). Regarding the MIC distribution of each association, aztreonam-relebactam appeared to be more efficient than aztreonam-vaborbactam, but both are less efficient than aztreonam-avibactam (Figure 1).

The addition of avibactam, relebactam or vaborbactam decreases the MIC of aztreonam by at least four dilutions for 100%, 93.5% and 90.3% of the strains, respectively (Table 1). The susceptibility rates of aztreonam-avibactam, aztreonam-vaborbactam and aztreonam-relebactam with a standard exposure of aztreonam (meaning 1g × 3, IV according to EUCAST guidelines) (≤1 mg/L for Enterobacterales and ≤0.001 mg/L for *S. maltophila* and *P. aeruginosa*) were 84.6% (44/52), 55.8% (29/52) and 34.6% (18/52) for Enterobacterales and 0% for all combinations for *P. aeruginosa* and *S. maltophila*, respectively. The susceptibility rates of aztreonam-avibactam, aztreonam-vaborbactam and aztreonam-relebactam with a high exposure of aztreonam (meaning 2g × 4, IV according to EUCAST guidelines) (≤4 mg/L for Enterobacterales and ≤16 mg/L for *S. maltophila* and *P. aeruginosa*) were 92.3% (48/52), 78.9% (41/52) and 57.7% (30/52) for Enterobacterales, 75% (4/5), 60% (3/5) and 60% (3/5) for *P. aeruginosa* and 100% (5/5), 100% (5/5) and 40% (2/5) for *S. maltophila* isolates, respectively (Table 1). As previously observed for the aztreonam-avibactam combination [5], the MICs of aztreonam-relebactam and aztreonam-vaborbactam were higher for NDM-producing *E. coli* compared to NDM-producing *K. pneumoniae*.

## 3. Discussion

In conclusion, as previously demonstrated for the aztreonam/ceftazidime-avibactam combination [10,11,12,13,14,15,16,17,18], aztreonam plus imipenem-relebactam [19,20,21] and aztreonam plus meropenem-vaborbactam [21] might be useful options to treat infections caused by aztreonam-non-susceptible MBL-producing Gram-negative strains. Despite the fact that avibactam seems to be the most potent inhibitor in association with aztreonam, it should be noted that ceftazidime is strongly hydrolyzed by ESBL and MBL with MICs >256 mg/L resulting in a non-residual effect of this drug in the tripartite combination aztreonam-ceftazidime-avibactam. Contrarily, imipenem and meropenem are not hydrolyzed by ESBL or AmpC and might retain partial activity on MBL (sometimes the MICs range from 2 to 8 mg/L). Accordingly, the tripartite associations of aztreonam-imipenem-relebactam and aztreonam-meropenem-vaborbactam might benefit not only the activity of the inhibitor with aztreonam, but also partly the residual activity of the carbapenem. Notably, our collection of Gram-negative bacteria included only a few isolates of *P. aeruginosa* and *S. maltophilia*. Due to this limitation, the results of the susceptibility of these two species have to be confirmed in further studies.

According to the recent guidelines of the Infectious Diseases Society of America (IDSA) [22] and the European Society of Clinical Microbiology and Infectious Diseases (ESCMID) [23], aztreonam-avibactam association produced by injecting aztreonam and ceftazidime-avibactam simultaneously [24] is considered the first-line treatment for infections caused by MBL producers. This is particularly the case for NDM producers that often display an increased MIC to cefiderocol [4,25,26,27], the other potential option for the treatment of infections caused by MBL producers [22,23].

In our collection, four NDM-producing *E. coli* displayed resistance to the aztreonam-avibactam association. In *E. coli*, the impact of the alteration of PBP3 has been recently described to be responsible for increased resistance to aztreonam-avibactam [11] and a cross-resistance to cefiderocol [28,29]. With the recent emergence of aztreonam-avibactam resistance among carbapenemase-producing Enterobacterales [11,29,30], these tripartite associations of aztreonam-imipenem-relebactam and aztreonam-meropenem-vaborbactam have to be taken into consideration as last-resort treatment options. Unfortunately, resistance to aztreonam-meropenem-vaborbactam has already been reported in a KPC variant producing *K. pneumoniae* [31]. Accordingly, the rapid development of additional options, such as new combinations with more potent inhibitors (e.g., zidebactam and taniborbactam) are mandatory [32,33,34]. However, as it was observed for aztreonam-avibactam, which is not already marketed, resistance to these very novel inhibitors already exists, as reported for cefepime-taniborbactam and NDM-9 producers [35,36].

In conclusion, aztreonam-vaborbactam and aztreonam-relebactam could be useful options for the treatment of infections caused by aztreonam-resistant MBL-producing isolates but with a potential lower efficiency compared to aztreonam-avibactam.

## 4. Materials and Methods

A total of 52 non-duplicate MBL-producing Enterobacterales, 5 MBL-producing *P. aeruginosa* and 5 multidrug-resistant *S. maltophila* isolates were used in this study. All strains were sequenced using the Illumina technique and the resistance genes were identified using Resfinder 4.1 (https://cge.food.dtu.dk/services/ResFinder/, 27 September 2023).

The MICs of aztreonam, meropenem-vaborbactam and imipenem-relebactam were determined by Etest^®^ (bioMérieux, La Balme-les-Grottes) according to EUCAST recommendations. For aztreonam-avibactam, aztreonam-vaborbactam and aztreonam-relebactam associations, MICs were determined using Etest^®^ on MH agar supplemented with 8 mg/L of avibactam (provided by Pfizer, France), 8 mg/L of vaborbactam (CliniSciences, Nanterre, France) and 4 mg/L of relebactam (provided by MSD, Puteaux, France). The MICs of aztreonam-avibactam, aztreonam-vaborbactam and aztreonam-relebactam were interpreted in the same way as aztreonam alone, according to EUCAST guidelines as updated in 2022 (Table 2).

Since no recommendation exists regarding these molecules for *S. maltophila*, interpretation criteria of *Pseudomonas* spp. were used (Table 3). A control of the accuracy of β-lactam inhibitor-supplemented MH agar was performed. For this purpose, the MICs of 13 strains, obtained with the imipenem and meropenem Etest^®^ (bioMérieux, France) on MH agar supplemented with 8 mg/L of vaborbactam and 4 mg/L of relebactam, were compared to the MICs obtained with the imipenem-relebactam and meropenem-vaborbactam Etest^®^ (bioMérieux, France). The difference in MICs obtained between the two methods did not exceed two dilutions, validating the methodology used in this study (Table 3). However, due to the relatively low number of tested isolates, the full validity of this method might have to be confirmed in further studies.

Of note is the fact that the accuracy of determining the MIC of aztreonam-avibactam using MH agar supplemented with 8 mg/L of avibactam has already been demonstrated on this strain collection. In the previous study, this method was also demonstrated to give similar results with the Etest^®^ strip superposition method [5]. *E. coli* ATCC 53126, *K. pneumoniae* ATCC 700,603 and two KPC-producing strains were used as quality controls.

## Figures and Tables

**Figure 1 antibiotics-12-01493-f001:**
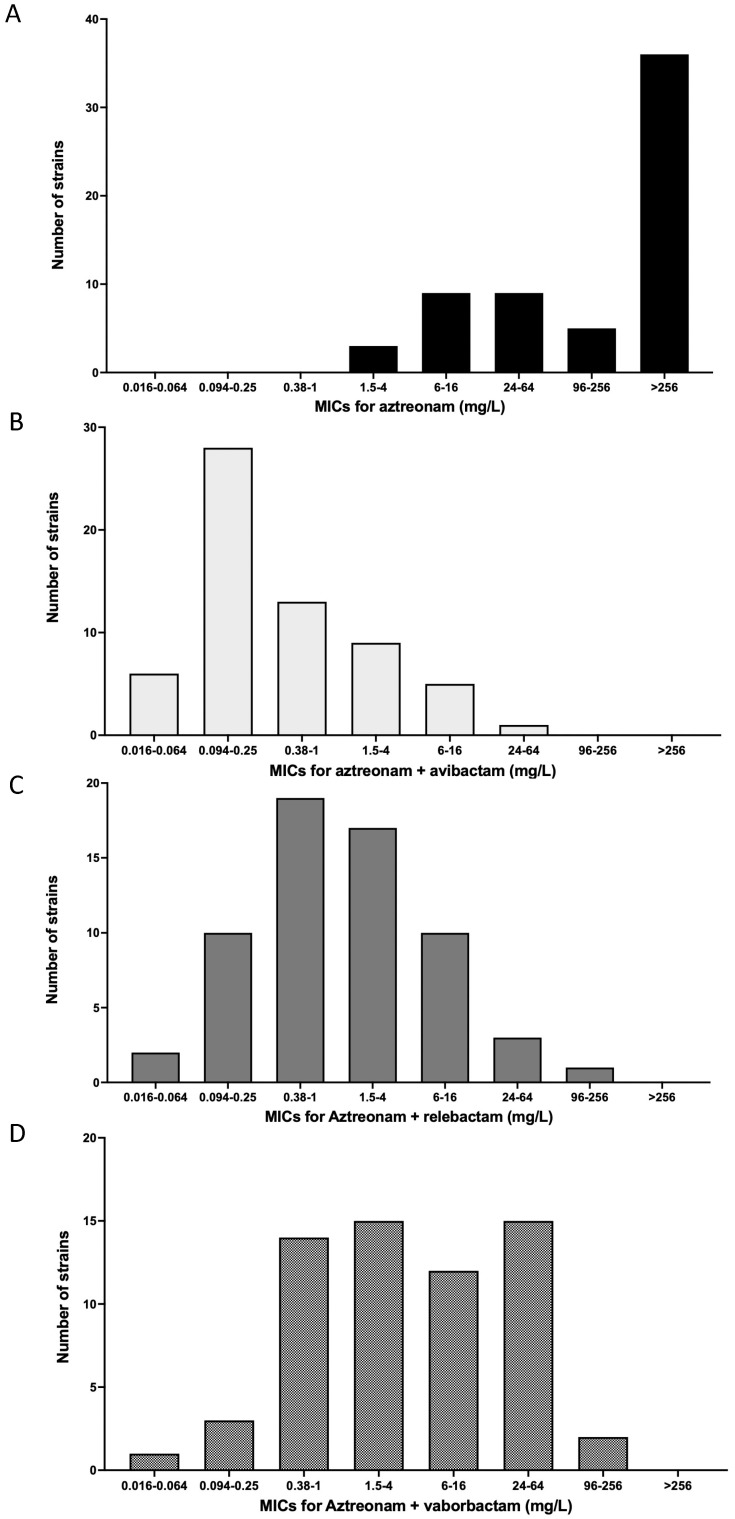
MICs of aztreonam (**A**), aztreonam-avibactam (**B**), aztreonam-relebactam (**C**) and aztreonam-vaborbactam (**D**).

**Table 1 antibiotics-12-01493-t001:** MICs of aztreonam (ATM), aztreonam-avibactam (ATM + AVI), aztreonam-relebactam (ATM + REL) and aztreonam-vaborbactam (ATM + VAB).

Species	β-Lactamases	MICs (mg/L)			
		ATM	ATM + AVI	AZM+ REL	ATM + VAB
*E. coli* ATCC 53126		0.047	ND	0.047	0.047
*K. pneumoniae* ATCC 700603		64	ND	0.5	2
*E. coli*	KPC	>256	ND	0.38	2
*K. pneumoniae*	KPC	>256	ND	0.125	0.25
*E. coli*	NDM-1 + OXA-1 + OXA-10 + CMY-16 + TEM-1	32	0.125	0.125	0.5
*E. coli*	NDM-1 + CTX-M-15 + TEM-1	>256	1	12	24
*E. coli*	NDM-1 + OXA-1 + OXA-2 + CTX-M-15 + TEM-1	>256	2	12	32
*E. coli*	NDM-1 + CTX-M-15 + TEM-1	>256	6	32	192
*E. coli*	NDM-4 + CTX-M-15 + OXA-1	>256	6	16	24
*E. coli*	NDM-4 + CTX-M-15 + CMY-6	>256	6	8	24
*E. coli*	NDM-5 + TEM-1 + CTX-M-15	>256	8	24	64
*E. coli*	NDM-6 + CTX-M-15 + OXA-1	>256	1	3	8
*E. coli*	NDM-7 + CTX-M-15	>256	4	12	32
*K. pneumoniae*	NDM-1 + CTX-M-15 + SHV-11 + OXA-1	>256	0.125	1	1.5
*K. pneumoniae*	NDM-1 + CTX-M-15 + CMY-4 + OXA-1	>256	0.75	4	48
*K. pneumoniae*	NDM-1 + CTX-M-15 + OXA-1 + OXA-9 + TEM-1 + SHV-28 + SHV-11	>256	0.25	4	12
*K. pneumoniae*	NDM-1 + OXA-1 + SHV-11	>256	0.047	0.094	0.094
*K. pneumoniae*	NDM-1 + OXA-1 + CTX-M-15 + TEM-1 + SHV-28 + OXA-9 + CMY-6	>256	0.047	0.75	0.75
*K. pneumoniae*	NDM-1 + TEM-1 + CTX-M-15 + SHV-12 + OXA-9	>256	0.125	1.5	1.5
*K. pneumoniae*	NDM-1 + TEM-1 + CTX-M-15 + SHV-12 + OXA-9	>256	0.125	1.5	12
*K. pneumoniae*	NDM-1 + TEM-1 + CTX-M-15 + SHV-11 + OXA-1	>256	0.064	0.75	0.75
*P. stuartii*	NDM-1 + OXA-1 + CMY-6 + TEM-1	8	0.032	0.016	0.032
*Salmonella enterica*	NDM-1 + CTX-M-15 + TEM-1 + OXA-1 + OXA-9 + OXA-10	>256	0.125	0.75	0.75
*E. coli*	VIM-1 + CTX-M-3	>256	0.125	1.5	12
*E. coli*	VIM-4 + CTX-M15	16	1.5	1.5	8
*K. pneumoniae*	VIM-1 + SHV-5	>256	0.25	3	24
*K. pneumoniae*	VIM-1 + SHV-12	>256	0.125	1	6
*K. pneumoniae*	VIM-1 + CTX-M-15	>256	0.19	0.125	0.125
*K. pneumoniae*	VIM-1 + SHV-5	16	0.19	1	1
*K. pneumoniae*	VIM-1 + TEM-1 + SHV-5	>256	0.25	1	6
*K. pneumoniae*	VIM-1 + SHV-5	>256	0.25	12	32
*K. pneumoniae*	VIM-1 + SHV-5	>256	0.125	1.5	8
*K. pneumoniae*	VIM-19 + CTX-M-3 + TEM-1 + SHV-1	6	0.047	0.032	0.094
*E. cloacae*	VIM-1 + SHV-70	256	0.094	0.5	2
*E. cloacae*	VIM-4 + CTX-M-15 + TEM-1 + SHV-31	64	1	0.5	4
*C. freundii*	VIM-2 + TEM-1 + CTX-M-15	16	0.25	3	1.5
*C. freundii*	VIM-2 + TEM-1 + OXA-9 + OXA-10	32	1.5	1.5	4
*E. coli*	IMP-8 + SHV -12	128	0.19	0.75	4
*K. pneumoniae*	IMP-1 + TEM-15	3	0.094	0.125	0.38
*K. pneumoniae*	IMP-1 + TEM-15 + CTX-M-15	2	0.094	0.125	0.38
*K. pneumoniae*	IMP-8 + SHV -12	>256	0.094	0.125	4
*E. cloacae*	IMP-8 + SHV-12	12	0.032	0.125	0.38
*S. marscecens*	IMP-11	4	0.5	0.125	0.38
*E. cloacae*	GIM-1 + CTX-M-15	12	0.5	0.19	1
*C. freundii*	TMB-1 + overexpressed cephalosporinase	64	0.125	0.19	0.75
*K. pneumoniae*	NDM-1 + OXA-181 + SHV-11 + TEM-1 + CTX-M-15 + OXA-1	64	0.094	0.5	0.5
*K. pneumoniae*	NDM-1 + OXA-181 + SHV-27 + CTX-M-15 + TEM-1 + OXA-1	128	0.25	1	1
*K. pneumoniae*	NDM-1 + OXA-181 + SHV-11 + CTX-M-15 + OXA-1	256	0.19	1	4
*K. pneumoniae*	NDM-1 + OXA-181 + SHV-11 + TEM-1 + CTX-M-15 + OXA-9	>256	0.19	0.75	16
*K. pneumoniae*	NDM-1 + OXA-181 + SHV-2 + CTX-M-15 + OXA-1	>256	0.125	0.75	1
*C. freundii*	NDM-1 + OXA-181 + OXA-1 + OXA-9 + OXA-10 + CTX-M-15 + TEM-1	>256	0.75	4	32
*E. coli*	NDM-1 + OXA-48 + CTX-M-15	32	0.094	0.5	1.5
*E. coli*	NDM-1 + OXA-48 + CTX-M-15	>256	0.75	6	16
*K. pneumoniae*	NDM-1 + OXA-232 + CTX-M-15	64	0.094	0.5	1.5
*E. coli*	NDM-1 + OXA-232 + CTX-M-15	>256	1	6	1.5
*E. coli*	NDM-5 + OXA-232 + CTX-M-15	>256	1	6	32
**Percentage of susceptible strains of Enterobacterales with standard exposure**		**0%**	**84.6%**	**55.8%**	**34.6%**
**Percentage of susceptible strains of Enterobacterales with high exposure**		**8.1%**	**92.3%**	**78.9%**	**57.7%**
**Percentage of resistant strains of Enterobacterales**		**91.9%**	**13.5%**	**21.1%**	**42.3%**
*S. maltophilia*		>256	0.5	1.5	1.5
*S. maltophilia*		>256	2	3	64
*S. maltophilia*		>256	0.5	1	4
*S. maltophilia*		>256	2	3	48
*S. maltophilia*		>256	1	1	32
*P. aeruginosa*	VIM-2 + overexpressed Cephalosporinase	32	8	32	24
*P. aeruginosa*	IMP-2 + overexpressed Cephalosporinase	6	1.5	2	2
*P. aeruginosa*	IMP-1 + overexpressed Cephalosporinase	24	3	4	16
*P. aeruginosa*	IMP-1 + overexpressed Cephalosporinase	96	32	128	96
*P. aeruginosa*	IMP-1 + overexpressed Cephalosporinase	12	3	8	6
**Percentage of susceptible strains with standard exposure of aztreonam in the combination (total)**		**0%**	**66.1%**	**45.2%**	**29.0%**
**Percentage of susceptible strains with high exposure of aztreonam in the combination (total)**		**4.8%**	**87.1%**	**79.1%**	**56.4%**
**Percentage of resistant strains (total)**		**95.2%**	**12.9%**	**21.0%**	**43.6%**
**Percentage of strains with aztreonam MIC reduction ≥ 4-fold dilution**			**100%**	**93.5%**	**90.3%**

Black: resistant; Grey: susceptible at high exposure; Blank: susceptible at standard dosage.

**Table 2 antibiotics-12-01493-t002:** EUCAST breakpoint of aztreonam for Enterobacterales and *Pseudomonas aeruginosa* (https://www.eucast.org/clinical_breakpoints, accessed on 27 September 2023).

	MICs (mg/L)	
	Enterobacterales	*P. aeruginosa*
Susceptible with standard exposure	≤1	≤0.001
Susceptible with high exposure	≤4	≤16
Resistant	>4	>16

**Table 3 antibiotics-12-01493-t003:** MICs of imipenem-relebactam (IMP + REL) and meropenem-vaborbactam (MEM + VAB) obtained using Etest^®^ and supplemented Mueller–Hinton agar.

		MICs			
		IMP+ REL Etest^®^	IMP+ REL Agar Method ^a^	MEM+VAB Etest^®^	MEM+VAB Agar Method ^b^
*E. coli*	NDM-1 + OXA-1 + OXA-2 + CTX-M-15 + TEM-1	4	8	6	8
*E. coli*	NDM-4 + CTX-M-15 + OXA-1	16	>32	32	>32
*E. coli*	NDM-5 + TEM-1 + CTX-M-15	>32	>32	>32	>32
*K. pneumoniae*	NDM-1 + TEM-1 + CTX-M-15 + SHV-12 + OXA-9	2	1	2	4
*P. stuartii*	NDM-1 + OXA-1 + CMY-6 + TEM-1	4	8	2	2
*E. coli*	VIM-4 + ESBL	6	4	3	1
*K. pneumoniae*	VIM-1 + SHV-5	3	4	1.5	2
*K. pneumoniae*	VIM-19 + CTX-M-3 + TEM-1 + SHV-1	1	2	0.75	1
*K. pneumoniae*	NDM-1 + OXA-181 + SHV-11 + CTX-M-15 + OXA-1	4	8	12	12
*E. coli*	NDM-1 + OXA-48 + ESBL	2	4	2	2
*E. coli*	NDM-5 + OXA-232 + ESBL	2	2	16	8
*S. maltophilia*		>32	>32	>32	>32
*P. aeruginosa*	IMP-1 + overexpressed Cephalosporinase	>32	16	>32	>32

^a^ Mueller–Hinton agar supplemented with 4 mg/L of relebactam. ^b^ Mueller–Hinton agar supplemented with 8 mg/L of vaborbactam.

## Data Availability

The data presented in this study are available on request from the corresponding author.

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
