# Peer review of "In Vitro Susceptibility of Aztreonam-Vaborbactam, Aztreonam-Relebactam and Aztreonam-Avibactam Associations against Metallo-β-Lactamase-Producing Gram-Negative Bacteria"

_antibiotics, 2023, doi:10.3390/antibiotics12101493_

Round 1

Reviewer 1 Report

The manuscript studies the effect of beta-lactamase inhibitors on the in-vitro activity of aztreonam. The conclusion (line 29 onwards) goes beyond the data by speculating that in-vivo application of aztreonam and marketed beta-lactam/beta-lactamase inhibitor combinations will result in improved coverage of strains producing metallo-beta-lactamases (MBLs).  The conclusion should be more cautiously worded.

In general compound adjectives such as "in-vitro", "first-line", "non-duplicate" and "Gram-negative" should be hyphenated. "Gram" should have an initial capital.

There are spurious plurals in several places in the text (e.g. line 62 "inhibitor", not "inhibitors").  Careful editing is required.

Line 51: the statement that MBLs do not hydrolyse aztreonam is incorrect, many MBLs slowly hydrolyse this agent.

Lines 58 and 132, it is a modified penicillin-binding protein 3 that confers resistance.  This should be spelt out in full the first time then shortened to PBP 3. 

Why is section 124-134 highlighted?

Basically OK, but needs systematic editing.

Author Response

Reviewer 1

The manuscript studies the effect of beta-lactamase inhibitors on the in-vitro activity of aztreonam. The conclusion (line 29 onwards) goes beyond the data by speculating that in-vivo application of aztreonam and marketed beta-lactam/beta-lactamase inhibitor combinations will result in improved coverage of strains producing metallo-beta-lactamases (MBLs).  The conclusion should be more cautiously worded.

Answer: It has been modified in the revised version of the manuscript : “As previously demonstrated for aztreonam / ceftazidime-avibactam combination, aztreonam plus imipenem-relebactam and aztreonam plus meropenem-vaborbactam might be useful options, but with potential less efficiency, to treat infections caused by aztreonam-non-susceptible MBL-producing Gram-negative strains..”

In general compound adjectives such as "in-vitro", "first-line", "non-duplicate" and "Gram-negative" should be hyphenated. "Gram" should have an initial capital.

Answer: It has been modified in the revised version of the manuscript

There are spurious plurals in several places in the text (e.g. line 62 "inhibitor", not "inhibitors").  Careful editing is required.

Answer: This manuscript has been carefully checked.

Line 51: the statement that MBLs do not hydrolyse aztreonam is incorrect, many MBLs slowly hydrolyse this agent.

Answer: The MBLs included in this study are the most frequently isolated worldwide and do not hydrolyze aztreonam. This clarification was added in the text : “Since current MBLs, such as VIM, NDM or IMP, are not able to hydrolyze aztreonam”

Lines 58 and 132, it is a modified penicillin-binding protein 3 that confers resistance.  This should be spelt out in full the first time then shortened to PBP 3. 

Answer: It has been modified in the revised version of the manuscript

Why is section 124-134 highlighted?

Answer: This is a formatting error, and it has been modified

Reviewer 2 Report

The authors of this manuscript describe their in vitro work with beta-lactamase combination with atreonam against MBL-producing gram negatives, Pseudomonas, and Stenotrophomonas isolates.  The following are comment that require author comments:

1.  Why only 5 isolates of Pseudomonas and Stenotrophomonas?  Authors need to add additional isolates or justify the scientific merits of 5 isolates to make drug efficacy claims. 

2.  Figure 1 needs additional captions on the x-axis for figure 1c.  may be appropriate for these MICs to be labelled a-d.  Additionally, a dotted line on the aztreonam breakpoint would be useful for the readers.  authors need to rectify. 

3.  Table 2 lists the EUCAST breakpoints for the isolates to aztreonam.  Authors need to examine these and ensure these are correct.  a reference would be useful for this table also. 

4.  Table 4 requires references to authenticate validity on this scientifically.  Are 12 isolates of enterobacterales enough to show the methods are valid?  is 1 isolate of stenotrophomonas and pseudomonas enough for scientific validity? 

5.  conclusions are lacking and authors need to summarize the results for a conclusion. 

6.  authors discuss cefiderocol therapy but did not include this antimicrobial in any of the in vitro work so I would remove this compound from the body of this paper. 

minor comments:

line 58 requires the abbreviation spelled out as the first time use.  Authors need to examine the entire manuscript for this. 

Need to ensure that all drugs are spelled correctly without hyphenation. 

Author Response

Reviewer 2

The authors of this manuscript describe their in vitro work with beta-lactamase combination with atreonam against MBL-producing gram negatives, Pseudomonas, and Stenotrophomonas isolates.  The following are comment that require author comments:

  1. Why only 5 isolates of Pseudomonas and Stenotrophomonas?  Authors need to add additional isolates or justify the scientific merits of 5 isolates to make drug efficacy claims. 

Answer: We tested these associations on the same collection of strains that we used for the evaluation of the association aztreonam-avibactam in a previous study (Emeraud et al, AAC 2018). We are aware that this is a limitation of this study and that we need to investigate these associations for a larger collection of non-fermenting bacilli. Accordingly, the following sentence has been added in the revised version of the manuscript : “Of note, our collection of Gram-negative included only few isolates of P. aeruginosa and S. maltophilia. Due to this limitation, the results of the susceptibility regarding these two species have to be confirmed in futher studies. »

  1. Figure 1 needs additional captions on the x-axis for figure 1c.  may be appropriate for these MICs to be labelled a-d.  Additionally, a dotted line on the aztreonam breakpoint would be useful for the readers.  authors need to rectify. 

Answer: The Figure 1 has been modified according to the reviewer’s recommendations. The breakpoints used in this study are those of aztreonam and are listed in the table 2. We decided not to show all these breakpoints on the figure because there is too much data (Pseudomonas/Enterobacterales, Standard exposure/high exposure) resulting in difficulties to read the figure.

  1. Table 2 lists the EUCAST breakpoints for the isolates to aztreonam.  Authors need to examine these and ensure these are correct.  a reference would be useful for this table also. 

Answer: These breakpoints have been checked on the EUCAST website and a reference has been added

  1. Table 3 requires references to authenticate validity on this scientifically.  Are 12 isolates of enterobacterales enough to show the methods are valid?  is 1 isolate of stenotrophomonas and pseudomonas enough for scientific validity? 

Answer: We agree that 12 Entrobacterales, 1 S. maltophilia and 1 P. aeruginosa might be not enough to definitively validate but we did not find any significant errors between the two methods with all the tested isolates despites different species and different mechanisms were tested. However, we acknowledge this limitation and we added the following sentence as limitation in the revised version of the manuscript : “However, due to the relatively low number of tested isolates, the full validity of this method might have to be confirm in further studies. »

  1. conclusions are lacking and authors need to summarize the results for a conclusion. 

Answer: a conclusion has been added : “In conclusion, aztreonam-vaborbactam and aztreonam-relebactam could be useful option for the treatment of infections caused by aztreonam-resistant MBL-producing isolates but with a potential less efficiency compared to aztreonam-avibactam”.

  1. authors discuss cefiderocol therapy but did not include this antimicrobial in any of the in vitro work so I would remove this compound from the body of this paper. 

Answer: We introduced cefiderocol in the introduction. It is true that this is not the main subject of this study but, in our opinion, it is crucial to specify that mutations in PBP3 confer cross resistance to aztreonam-avibactam and cefiderocol, the two sole options included in all consensus for the treatment of infections caused by MBL-producers.

minor comments:

line 58 requires the abbreviation spelled out as the first time use.  Authors need to examine the entire manuscript for this.

Answer: The abbreviation PBP3 has been introduced in the revised version of the manuscript

Reviewer 3 Report

In the Manuscript ID: antibiotics-2573686, entitled ‘’In vitro susceptibility of aztreonam-vaborbactam, aztreonam-relebactam and aztreonam-avibactam associations against metallo- β-lactamase-(MBL) producing Gram-negative bacteria by Cécile EMERAUD et al., the authors present data on the antimicrobial susceptibility testing to novel β-lactams/ β-lactamase inhibitor combinations of a sample of metallo-β-lactamase-producing Gram-negative bacteria. This is an interesting paper that addresses the difficulties in the effective treatment of MBL producers.  The introduction is appropriate, the materials and methods section the presentation of the results may be improved. The discussion and the conclusions are consistent with the results. The references are appropriate. Minor grammatical errors need to be corrected.

Major comments:

-          In the Materials and Methods, the authors should describe the methods used for characterization of the β-lactamase content of the isolates.

-          In Table 3, the term ESBL is used but the extended-spectrum β-lactamase is not defined, whereas in some isolates the ESBL (e.g. CTX-M- and SHV- type) is characterized. Please, provide a consistent presentation of the results, i.e using the term ESBL or the characterization of the ESBL type where is missing.

Minor comments:

-          Lines 15, 16, 17 28: Please use italics for the species name.

-          Lines 37-38: ‘’Gram negative’’ please replace as ’’Gram negative’’

-          Line 46: ‘’MBLs-producers ‘’please replace as ’’MBL producers’’

-          Line 48: ‘’MBL hydrolyzing activity’’please replace as ’’MBL-hydrolyzing activity,’’

-          Line 50: ’’MBLs’’ and ‘’2 mg/L’’ please replace as ’’MBLs,’’ and ‘’2 mg/L,’’

-          Line 63:  ’’KPC enzymes’’ please replace as ’’KPC enzymes,’’

-          Line 98: ‘’≤ 0.001’’ please replace as ‘’≤ 0.001mg/L’’

-          Line 103: ‘’≤ 16’’ please replace as‘’≤ 16 mg/L’’

-          Line 112: ‘’To concluded’’ you may use in conclusion, conclusively

-          Line 116: ‘’note’’ please replace as ‘’noted’’

-          Line 117:  ‘’not’’ please replace as  ‘’non’’

-          Line 122:  ‘’aztreonam’’ please replace as ‘’aztreonam,’’

-          Line 135: ‘’ association’’ please replace as ‘’ associations’’

-          Lines 139-140: ‘’the rapid development a clinical availability of other options‘’ please rephrase for clarity

-          Table 1: The cells in the lines below are shaded and the percentages are not shown.

Percentage of susceptibles Enterobacterales with standard exposure

Percentage of susceptibles Enterobacterales with high exposure

Percentage of resistants Enterobacterales

Moreover, correct as:

Percentage of susceptible strains of Enterobacterales with standard exposure

Percentage of susceptible strains of  Enterobacterales with high exposure

Percentage of resistant strains of Enterobacterales

Percentage of susceptible strains withstandard exposure of aztreonam in the combination (total)

Percentage of susceptible strains with high exposure of aztreonam in the combination (total)

Percentage of resistant strains (total)

-        Table 2: please provide the breakpoints when available for the remaining antimicrobial agents tested

Minor editing of English language required

Author Response

Reviewer 3

In the Manuscript ID: antibiotics-2573686, entitled ‘’In vitro susceptibility of aztreonam-vaborbactam, aztreonam-relebactam and aztreonam-avibactam associations against metallo- β-lactamase-(MBL) producing Gram-negative bacteria by Cécile EMERAUD et al., the authors present data on the antimicrobial susceptibility testing to novel β-lactams/ β-lactamase inhibitor combinations of a sample of metallo-β-lactamase-producing Gram-negative bacteria. This is an interesting paper that addresses the difficulties in the effective treatment of MBL producers.  The introduction is appropriate, the materials and methods section the presentation of the results may be improved. The discussion and the conclusions are consistent with the results. The references are appropriate. Minor grammatical errors need to be corrected.

Major comments:

-          In the Materials and Methods, the authors should describe the methods used for characterization of the β-lactamase content of the isolates.

Answer: More details about the methods used to characterize the collection have been added in the revised version of the manuscript. “ All strains were sequenced using the Illumina technique and the resistance genes were identified using Resfinder 4.1 (https://cge.food.dtu.dk/services/ResFinder/)”

-          In Table 3, the term ESBL is used but the extended-spectrum β-lactamase is not defined, whereas in some isolates the ESBL (e.g. CTX-M- and SHV- type) is characterized. Please, provide a consistent presentation of the results, i.e using the term ESBL or the characterization of the ESBL type where is missing.

Answer: The ESBL were characterized and the term ESBL has been replace by the corresponding enzyme name in the revised version of the manuscript

Minor comments:

-          Lines 15, 16, 17 28: Please use italics for the species name.

Answer: It has been done accordingly

-          Lines 37-38: ‘’Gram negative’’ please replace as ’’Gram negative’’

Answer: It has been done accordingly

-          Line 46: ‘’MBLs-producers ‘’please replace as ’’MBL producers’’

Answer: It has been done accordingly

-          Line 48: ‘’MBL hydrolyzing activity’’please replace as ’’MBL-hydrolyzing activity,’’

Answer: It has been done accordingly

-          Line 50: ’’MBLs’’ and ‘’2 mg/L’’ please replace as ’’MBLs,’’ and ‘’2 mg/L,’’

Answer: It has been done accordingly

-          Line 63:  ’’KPC enzymes’’ please replace as ’’KPC enzymes,’’

Answer: It has been done accordingly

-          Line 98: ‘’≤ 0.001’’ please replace as ‘’≤ 0.001mg/L’’

Answer: It has been done accordingly

-          Line 103: ‘’≤ 16’’ please replace as‘’≤ 16 mg/L’’

Answer: It has been done accordingly

-          Line 112: ‘’To concluded’’ you may use in conclusion, conclusively

Answer: It has been done accordingly

-          Line 116: ‘’note’’ please replace as ‘’noted’’

Answer: It has been done accordingly

-          Line 117:  ‘’not’’ please replace as  ‘’non’’

Answer: It has been done accordingly

-          Line 122:  ‘’aztreonam’’ please replace as ‘’aztreonam,’’

Answer: It has been done accordingly

-          Line 135: ‘’ association’’ please replace as ‘’ associations’’

Answer: It has been done accordingly

-          Lines 139-140: ‘’the rapid development a clinical availability of other options‘’ please rephrase for clarity

Answer: It has been rephrased as follow: “Accordingly, the rapid development additional options such as new combinations with more potent inhibitors (zidebactam, taniborbactam) is mandatory » 

-          Table 1: The cells in the lines below are shaded and the percentages are not shown.

Percentage of susceptibles Enterobacterales with standard exposure

Percentage of susceptibles Enterobacterales with high exposure

Percentage of resistants Enterobacterales

Answer: It has been corrected accordingly

Moreover, correct as:

Percentage of susceptible strains of Enterobacterales with standard exposure

Percentage of susceptible strains of  Enterobacterales with high exposure

Percentage of resistant strains of Enterobacterales

Percentage of susceptible strains withstandard exposure of aztreonam in the combination (total)

Percentage of susceptible strains with high exposure of aztreonam in the combination (total)

Percentage of resistant strains (total)

 Answer: All these minor comments were considered for the revised version of the manuscript

-        Table 2: please provide the breakpoints when available for the remaining antimicrobial agents tested

 Answer: MICs of aztreonam-avibactam, aztreonam-vaborbactam and aztreonam-relebactam were interpreted like aztreonam alone according to EUCAST guidelines as updated in 2022. This sentence has been change in the revised version of the manuscript for clarification.

Round 2

Reviewer 2 Report

agree with the changes the authors have made.  I have no further concerns.